# Immunohistochemical Expression of Glutathione Peroxidase-2 (Gpx-2) and Its Clinical Relevance in Colon Adenocarcinoma Patients

**DOI:** 10.3390/ijms241914650

**Published:** 2023-09-27

**Authors:** Marlena Brzozowa-Zasada, Angela Ianaro, Adam Piecuch, Marek Michalski, Natalia Matysiak, Katarzyna Stęplewska

**Affiliations:** 1Department of Histology and Cell Pathology in Zabrze, Faculty of Medical Sciences in Zabrze, Medical University of Silesia in Katowice, 40-055 Katowice, Poland; 2Department of Pharmacy, School of Medicine, University of Naples Federico II, Via D. Montesano 49, 80131 Naples, Italy; 3Department of Pathology, Institute of Medical Sciences, University of Opole, 45-052 Opole, Poland

**Keywords:** glutathione peroxidase, immunohistochemistry, prognostic marker, colon adenocarcinoma, oxidative stress, immunogold labelling, 5-year survival

## Abstract

Glutathione peroxidase 2 (Gpx-2) is a selenoenzyme with antioxidant capabilities that may play a role in cancer development. Hence, we investigated the immunohistochemical expression of Gpx-2 protein in colon adenocarcinoma samples derived from patients with colon adenocarcinoma who did not receive any form of treatment prior to the surgical procedure. The associations between the immunohistochemical expression of Gpx-2 and clinical parameters were analysed using the Chi^2^ test and Fisher’s exact test. A Kaplan–Meier analysis and the log-rank test were used to verify the relationship between the intensity of Gpx-2 expression and the 5-year survival rate of patients. In total, 101 (80.80%) samples had strong Gpx-2 protein expression and 24 (19.20%) samples were characterized with low expression. The high expression of Gpx-2 was correlated with the histological grade of the tumour (*p* < 0.001), PCNA immunohistochemical expression (*p* < 0.001), depth of invasion (*p* = 0.001) and angioinvasion (*p* < 0.001). We can conclude that high expression of Gpx-2 is correlated with reduced survival of colon adenocarcinoma patients (log-rank, *p* < 0.001).

## 1. Introduction

Colon adenocarcinoma (COAD) is one of the most common types of colorectal cancer (CRC) and has become more prevalent in recent years. It is a heterogeneous malignancy with a high recurrence probability and dismal prognosis. Currently, effective therapeutic strategies against recurrence and metastatic COAD remain rare [1]. Therefore, it is important to develop a new prognostic tool to identify patients at high risk of recurrence who require more attention and treatment. The chances of patients surviving depend on the stage at which COAD is diagnosed. Individuals diagnosed at a metastatic stage have a lower chance of survival [2]. For those diagnosed with cancer at an early stage, the 5-year survival rate is 90 percent, but it drops to only 13 percent for those diagnosed at a later stage [3,4].

It should be pointed out that in both research and clinical practice, adenocarcinomas of the colon and rectum are often grouped as a single cancer called colorectal cancer (CRC). This is because they both develop in the large bowel. However, despite similarities in anatomy, function and histology, it is important to note that there are differences in molecular carcinogenesis, pathology, surgical procedures and treatment between tumours localized in the colon and rectum [5,6].

The glutathione peroxidases family consists of different isozymes (Gpx-1-8) that help in reducing H_2_O_2_ or organic hydroperoxides to water or corresponding alcohols [7,8]. Gpx-2, in particular, is an enzyme that contains selenium and is found mainly in the gastrointestinal tract. It functions as an antioxidant by scavenging reactive oxygen species (ROS) and has drawn attention to its role in cancer formation and progression. Studies have shown that overexpression of Gpx-2 contributes to the initiation, development and spread of lung, hepatocellular and bladder cancers [9,10,11]. Additionally, Lei et al. suggested that Gpx-2 plays a role in forming a comprehensive multigene prognosis model for glioblastoma multiforme [12].

As mentioned above, Gpx-2 is found in cells of the gastrointestinal tract, particularly in epithelial cells at the base of the proliferating crypt [13,14]. It is also highly expressed in adenomas and carcinomas within the gastrointestinal tract, suggesting a potential role in the regulation of redox homeostasis during carcinogenesis [15]. Unfortunately, there is a lack of data on the immunohistochemical expression of Gpx-2 in colon and rectal adenocarcinoma tissues. There is also a lack of information on its prognostic activity in patients with this type of tumour. Muravaki et al. found that selenoproteins such as Gpx-1 and Gpx-3, which are highly susceptible to degradation during selenium depletion, showed a marked decrease in expression levels in cancer tissues within the large bowel. Interestingly, Gpx-2, which is resistant to selenium depletion, showed an increase in expression [16]. In colorectal cancer cell lines, Gpx-2 silencing led to the accumulation of ROS and increased sensitivity to H_2_O_2_-induced apoptosis. It also significantly reduced the ability of the cells to metastasise. The loss of clonogenic capacity was reversed with ROS neutralizing. Remarkably, cells lacking Gpx-2 exhibited a lack of differentiation capacity and developed slow-growing, undifferentiated tumours [9].

Our research aimed to investigate the expression of Gpx-2 protein in patients diagnosed with colon adenocarcinoma who had not received any treatment prior to surgery. We focused on European (Polish) populations, as there is no evidence of Gpx-2 protein expression in this group. We also investigated the correlation between Gpx-2 protein expression and clinicopathological factors in patients with colon adenocarcinoma, e.g., a correlation between Gpx-2 expression and proliferating cell nuclear antigen (PCNA) expression. In addition, our study aimed to evaluate the prognostic potential of Gpx-2 protein with respect to patient survival over a 5-year period. We also examined the intracellular distribution of Gpx-2 in cancer tissues using immunogold labelling.

## 2. Results

### 2.1. Patients’ Characteristics

The colon adenocarcinoma specimens included a cohort of 65 male and 60 female patients, with a median age of 65 years and an interquartile range of 56 to 77 years. In 50.40% of cases, tumours were located in the proximal colon, while 49.60% of cases were located in the distal part of the colon. Histological differentiation was divided into three levels to determine grading, with G1 comprising 22 cases (17.60%), G2 comprising 64 cases (51.20%) and G3 comprising 39 cases (31.20%), as shown in Table 1.

Within the study cohort, 101 colon adenocarcinoma samples showed robust expression of the Gpx-2 protein, representing 80.80% of the cohort. Conversely, only 24 samples showed decreased immunoreactivity, representing 19.20% of the total samples (Figure 1).

### 2.2. Correlations between Gpx-2 Immunohistochemical Expression and Clinicopathological Parameters (as Independent Variables)

By performing an immunohistochemical analysis, we were able to establish a correlation between the GPx-2 and immunohistochemical status of the patients and their clinicopathological characteristics, as well as their 5-year survival rate. The observed increase in the expression levels of Gpx-2 showed a significant correlation with the histological grade of the tumour as determined with the Chi^2^ test (*p* < 0.001). The results of this study indicate a substantial presence of Gpx-2 protein in tumours that were graded G1, G2 and G3. Specifically, 27.27%, 89.06% and 97.44% of tumours in these grades showed high levels of Gpx-2 protein expression. In contrast, Gpx-2 protein showed a marked decrease in immunohistochemical expression, with 16 (72.73%), 7 (10.94%) and 1 (2.56%) tumours showing low levels of expression in G1, G2 and G3, respectively. In addition, GPx-2 expression was found to be closely correlated with PCNA antigen expression (*p*-value < 0.001, Fisher’s exact test). Significant upregulation of Gpx-2 protein was observed in 29.41% and 88.89% of samples with low and high levels of PCNA immunoreactivity, respectively. The analysis seems to be interesting when both the PCNA and the Gpx-2 are treated as dependent variables. The number of subjects with high levels of PCNA and Gpx-2 expression was statistically higher (76.80% of the total group), as was the number of subjects with low levels of both PCNA and Gpx-2 expression (9.60% of the total group). Conversely, the number of patients with low levels of PCNA and high levels of Gpx-2 expression (4% of the total) was similar to those with high levels of PCNA and low levels of Gpx-2 expression (9.6% of the total) (Table 2).

It should also be noted that the expression of Gpx-2 showed a significant association with angioinvasion (*p* < 0.001, Fisher’s exact test). Of the patients with positive angioinvasion, 89 (90.82%) had high Gpx-2 immunohistochemical expression and 9 (9.18%) had low immunoreactivity. Conversely, 12 (44.44%) patients without angioinvasion showed strong Gpx-2 expression, while low Gpx-2 immunoreactivity was detected in 15 (55.56%) patients. The immunohistochemical expression of Gpx-2 was also associated with the depth of invasion (*p* = 0.001, Chi^2^ test). In patients classified as T1/T2, a high level of the immunohistochemical reaction was observed in 13 (54.17%) and a low level of expression in 11 (45.83%). For T3/T4 patients, a strong Gpx-2 immunohistochemical reaction was reported in 88 (87.13%) patients, while low expression was detected in 13 (12.87%). In the cohort of individuals diagnosed with stage I, 7 (50.00%) showed increased Gpx-2 expression levels and the same number of patients demonstrated a high level of Gpx-2 expression. Among patients diagnosed with stage II disease, about 90.32% were characterized with high levels of immunoreactivity, while only 15.63% showed low levels. In contrast, among patients diagnosed with stage III disease, 66 (82.50%) showed strong immunoreactivity, while 14 (17.50%) showed low levels of immunoreactivity. This finding was found to be statistically significant (*p* = 0.005, Chi^2^ test) (Table 3).

### 2.3. Prognostic Role of Gpx-2 Expression in Colon Adenocarcinoma

The aim of the present study was to investigate the prognostic relevance of Gpx-2 expression in patients with colorectal adenocarcinoma, with particular emphasis on its association with 5-year survival. Kaplan–Meier survival curves were used to evaluate all samples. The group of patients with decreased Gpx-2 expression showed a significantly improved 5-year survival compared to the patients with the high expression of this protein (log-rank, *p* < 0.001) (Figure 2).

The significance of Gpx-2 expression in relation to 5-year survival was also assessed in subgroups of patients stratified by grade of histological differentiation, depth of invasion, staging and PCNA expression. Specifically, the expression of Gpx-2 did not display any significant correlation with the 5-year survival of patients who were placed in G1 (log-rank test, *p* = 0.400), G2 (log-rank test, *p* = 0.385) and G3 (log-rank test, *p* = 0.068). Notably, within the cohort of patients with T1/T2-depth invasion, those with a reduced level of Gpx-2 immunohistochemical reactivity demonstrated significantly superior 5-year survival rates compared to their counterparts with elevated levels of this particular protein (log-rank test, *p* = 0.011). Comparable results were observed in patients with T3/T4 invasion depths (log-rank test, *p* = 0.008). The high expression of Gpx-2 in patients with stage I of the disease was connected with worse survival (log-rank test, *p* = 0.005). Similar results were obtained in patients with stage II and III colon adenocarcinoma (log-rank test, *p* = 0.012 versus *p* = 0.027).

Gpx-2 expression showed a significant correlation with 5-year survival in patients with elevated PCNA expression. Significant statistical results were observed in the 5-year survival of patients with elevated levels of PCNA and decreased Gpx-2 expression (log-rank test, *p* = 0.006) (Figure 3).

The results of the study suggest that several factors may have significant prognostic value, including Gpx-2 protein level, histological differentiation grade, invasion depth, angioinvasion and PCNA expression. Univariate Cox regression analyses were performed to determine the statistical significance of these variables. The results of our cohort study indicate that in patients with colorectal adenocarcinoma, histological differentiation grade and Gpx-2 expression were identified as independent indicators significantly associated with 5-year survival as demonstrated with the use of a multivariate analysis (Table 4).

### 2.4. Immunofluorescence Staining

We aimed to study the expression of Gpx-2 in colon adenocarcinomas through immunofluorescence. We selected 50 randomly chosen slides of tissue sections treated with an anti-Gpx2 antibody and Dako Liquid Permanent Red chromogen (LPR). This selection included 10 control samples, 25 samples with low expression identified using immunohistochemistry and 25 samples with high expression. Although we used this method as an additional measure, the results were encouraging, suggesting that LPR chromogen treatment of anti-Gpx-2-antibody-stained tissue sections could benefit immunofluorescence studies. Zeiss Zen 3.4 (blue edition) software was used to measure Gpx-2 expression levels in both normal and cancerous tissues. Both non-cancerous and cancerous mucosa cells showed a fluorescent signal that varied in intensity and was coloured red. In some cancer cells, the fluorescent signal was detected in the apical cytoplasmic regions, while in others, strong fluorescence was observed throughout the cytoplasm of the cells (Figure 4).

### 2.5. Intracellular Localization of Gpx-2 using the Method of Immunogold Labelling with the Use of Transmission Electron Microscopy (TEM)

The present study used immunogold labelling to reveal the subcellular distribution of the Gpx-2 protein in colon adenocarcinoma samples. Electron-dense black granules were observed within both cancerous and stromal cells, particularly within fibroblasts. In the cytoplasm, near the endoplasmic reticulum and mitochondria, electron-dense granules conjugated to an anti-Gpx-2 antibody were also visible. In addition, in cancer cells, small black granules were observed inside tiny vesicles in close proximity to the nuclear envelope. In the context of fibroblasts, these granules were distributed throughout the extended cytoplasm and also within the cisterns that make up the rough endoplasmic reticulum. In addition, electron-dense black granules were observed in close proximity to the mitochondria in cancer cells and stromal fibroblasts (Figure 5).

## 3. Discussion

A number of studies have shown that the progression of epithelial tumours is favoured by an elevated level of ROS. An enhanced level of ROS that is usually detected in cancerous cells may be due to several mechanisms, e.g., increased metabolic rate, mitochondrial dysfunction or changes in enzymes associated with ROS metabolic pathways [17,18,19]. It is widely recognized that a balance is necessary between the rate of ROS production and the elimination of free radical damage by the body’s antioxidant defence system. GPx-2 is a type of antioxidant enzyme that contains selenium and is found in the cells of the gastrointestinal tract. It is known to be responsible for scavenging H_2_O_2_ at the cellular level through glutathione-dependent mechanisms, thereby helping to prevent oxidative stress [20,21].

Results of our study demonstrated that expression of Gpx-2 in colon adenocarcinoma tissue was upregulated in comparison to that observed in the non-pathological tissue of the surgical margin. It is important to note that among our group of patients, 81% of colon adenocarcinoma samples displayed high levels of Gpx-2 protein expression, while only 19% exhibited low levels of immunoreactivity. We confirmed the presence of Gpx-2 in the cytoplasm of tumour cells using the immunogold labelling method. The protein was mainly found near membranous organelles such as the endoplasmic reticulum and mitochondria. It was noticed that Gpx-2 was present in not only cancer cells but also stromal cells, particularly in cancer-associated fibroblasts (CAFs).

We observed a significant correlation between high Gpx-2 expression and factors such as the histological grade of the tumour (*p* < 0.001, Chi^2^ test), depth of invasion (*p* = 0.001, Fisher’s exact test), angioinvasion (*p* < 0.001, Fisher’s exact test) and PCNA immunohistochemical expression (*p* < 0.001, Fisher’s exact test). Notably, we identified strong expression of Gpx-2 protein in 27% of G1 tumours, 89% of G2 tumours and 98% of G3 tumours. These findings suggest that Gpx-2 may play a crucial role in the progression of colon adenocarcinoma and could potentially be used as a biomarker for identifying patients with a more aggressive form of this cancer. Gpx-2 might be an important predictor for the prognosis of colon adenocarcinoma and a potential target for intervention and treatment of such patients. It should be pointed out that the expression of Gpx-2 was also linked to PCNA immunohistochemical expression (*p* < 0.001, Fisher’s exact test). Samples with low PCNA expression showed a high level of Gpx-2 reactivity in 29% of cases, while 89% of samples with high PCNA expression had high Gpx-2 reactivity. In this context, it should be pointed out that PCNA is a nuclear protein with a molecular mass of 36 kDa and is a specific marker of cell division. It is associated with DNA polymerase and synthesized shortly before the S-phase of the cell cycle [22]. It is noteworthy that patients with high Gpx-2 expression also had the most intense expression of PCNA. During the planning of our research, we also decided to examine the value of Gpx-2 expression in terms of 5-year survival in a cohort of patients who were stratified according to low and high levels of expression of PCNA. As mentioned above, our results demonstrated that the 5-year survival was significantly reduced in patients with high Gpx-2 expression and high PCNA expression. This finding may have clinical utility. Patients who show the high immunohistochemical expression of Gpx-2 and PCNA were also characterized with poor clinical outcomes. These patients should be treated as a special risk group. They may need to be treated differently after colon adenocarcinoma resection.

The results of our study are similar to those obtained by other authors. The high expression of Gpx-2 was also detected in the case of gastric, oesophageal, liver and breast cancers [23,24,25,26]. The high expression of Gpx-2 is associated with the growth [27] metastasis [9] and drug resistance [28] of cancer cells, as well as a low patient survival rate [23,24,25,26]. However, the opposite trend was observed in breast cancer, oesophageal squamous cell carcinoma and bladder cancer [25,29,30]. Peng et al. revealed that in non-small cell lung cancer cells (NSCLC), overexpression of Gpx-2 promoted epithelial–mesenchymal transition (EMT), migration and invasion whereas the knockdown of Gpx-2 showed the opposite effects and inhibited the metastasis of cancer cells in nude mice. Moreover, Gpx-2 was associated with a reduction in ROS accumulation and activation of the Vimentin and Snail proteins [31]. The protein Gpx-2 plays a vital role in inducing apoptosis in hepatocellular carcinoma cells (HCC cells) through lenvatinib therapy and serves as a biomarker for guiding treatment in HCC patients. Gpx-2 is a downstream gene regulated by β-catenin. The lenvatinib prevents the nuclear translocation of β-catenin, leading to the inhibition of Gpx-2 expression in HCC cells. Notably, in patients who received lenvatinib therapy, the Gpx-2 expression was correlated with the response to treatment. Patients with low expression of this protein had an objectively better prognosis than those with high expression of Gpx-2 [32]. The downregulation of β-catenin, Vimentin and Snail and the upregulation of E-cadherin were observed as a result of Gpx-2 silencing in pancreatic cancer cells. This led to the suppression of PC cell proliferation, metastasis and invasion [33]. The results of these in vitro experiments clearly showed that Gpx-2 may be involved in cancer progression, probably by activating EMT and signalling pathways associated with this process. This may have clinical implications. Drugs that have the potential to be inhibitors of this protein may be promising targets for cancer therapy.

In this place, it should be pointed out that the role of Gpx-2 in cancer development varies at different stages of the disease. In the initiation phase, Gpx-2 is involved in the defence mechanism and elimination of ROS. In the promotion phase, Gpx-2 may have a function to promote proliferation and prevent the elimination of malignant cells by inhibiting apoptotic cell death [34,35]. In the context of adenocarcinomas, it should be pointed out that GPx-2 may have a regulatory role in these malignancies, defending their highly proliferative cancer cells from p53-dependent oxidative damage. This phenomenon may be a common mechanism with which Gpx-2 promotes malignancy in adenocarcinoma-derived tumours that are susceptible to oxidative stress [35,36].

## 4. Materials and Methods

### 4.1. Patients and Tumour Samples

In the current study, samples of colon tissue were obtained from patients with established colon adenocarcinoma, confirmed with histopathological examination, who underwent colon resection procedures at Jaworzno Municipal Hospital between January 2014 and December 2015. Patients who received preoperative radiotherapy or chemotherapy, had severe medical problems or distant metastases, underwent resection for tumour recurrence, had inflammatory-bowel-disease-related colorectal cancer or had a histopathological subtype other than colon adenocarcinoma were excluded from the study.

According to a standardized protocol, histopathological sections were obtained from each surgical specimen, consisting of tumour fragments and parts of adjacent tissue without tumour abnormalities. The specimens were fixed in formalin and embedded in paraffin blocks. Paraffin blocks were then cut and sections were routinely H&E stained for the histopathological diagnosis. Marginal tissue sections were also examined. If any cancer cells were detected, the material was excluded from the study. To evaluate the prognostic significance of Gpx-2 protein, patients were followed for 5 years to evaluate 5-year survival.

### 4.2. Immunohistochemical and Immunofluorescence Staining

Paraffin-embedded tissue blocks with formalin-fixed colon adenocarcinoma specimens and resected margins were cut into 4 µm thick sections, fixed on Polysine slides and deparaffinized in xylene and rehydrated through a graded series of alcohol. To retrieve the antigenicity, the tissue sections were treated with microwaves in a 10 mM citrate buffer (pH 6.0) for 8 min each. Subsequently, sections were incubated with antibodies to Gpx-2 (GeneTex, polyclonal antibody, Cat. No. GTX100292, final dilution—1:800, Irvine, CA, USA) and PCNA (GeneTex, polyclonal antibody, Cat. No. GTX100539, final dilution—1:600, Irvine, CA, USA). For visualisation of protein expression, the sections were treated with a BrightVision (Cat. No. DPVB55HRP WellMed BV, ’t Holland 31, 6921 GX Duiven, the Netherlands) detected system and Permanent AP Red Chromogen (Dako LPR from Agilent Technologies Code K0640, Santa Clara, CA, USA). Mayer’s haematoxylin was used to counterstain the nuclei. We analysed sections of healthy mucosa from patients undergoing screening colonoscopy with no inflammatory or cancerous lesions to study the expression of Gpx-2 and PCNA. To analyse the results of immune histochemical staining, we followed the immunoreactive score used in previous publications [37,38,39]. The score was based on both the intensity and the number of cells with a positive immunohistochemical reaction, and it determined the presence of Gpx-2. The intensity was graded as follows: 0 for no signal, 1 for weak, 2 for moderate and 3 for strong staining. We assessed the frequency of positive cells semiquantitatively by evaluating the entire section, and each sample was scored on a scale of 0 to 4: 0 for negative, 1 for positive staining in 10–25% of cells, 2 for 26–50% of cells, 3 for 51–75% of cells and 4 for 76–100% of cells. We then calculated a total score of 0–12 and graded it as follows: I for scores 0–1, II for scores 2–4, III for scores 5–8 and IV for scores 9–12. We considered grade I as negative, and grades II, III and IV as positive. Grades I and II represented low expression (no or weak staining), and grades III and IV represented high expression (strong staining). The evaluation was performed by two independent pathologists, and any differences were resolved with a consensus.

Additionally, tissue sections treated with an anti-Gpx-2 antibody and Dako Liquid Permanent Red (LPR) were visualized with a confocal fluorescent microscope (Zeiss LSM 980 with Airscan 2; Zeiss; Jena, Germany). LPR fluorescence representing Gpx-2 protein was visualized with 592 nm excitation and 574–735 nm emission using TexRed filter sets. The intensity of Gpx-2 expression in both non-neoplastic tissue and tumour tissue was determined using the software Zeiss Zen 3.4 (blue edition) version 3.4.91.00000 (Zeiss; Germany).

### 4.3. Immunogold Electron Microscopy

In the present study, 15 samples of colon adenocarcinoma and colon tissues without any pathological changes were immobilized in a 4% solution of paraformaldehyde in 0.1 M phosphate-buffered saline (PBS) at room temperature for 2 h, followed by several rinses in PBS. Following washing, the specimens were dehydrated in a graded ethanol series and infiltrated for 30 min on ice in a 2:1 (*v*:*v*) ethanol/LR White mixture and a 1:2 (*v*:*v*) mixture. After that, the samples were infiltrated with pure LR White. An RMC Boeckeler Power Tomo PC ultramicrotome with a diamond blade (45°; Diatom AG, Biel, Switzerland) was used to cut ultrathin sections (70 nm). Ultrasections were immunolabeled and placed on 200 Formvar-coated mesh nickel grids. Sections on the grids were pre-incubated for 30 min by floating on drops of 50 mM NH4 Cl in PBS, followed by 30 min of blocking on drops of 1% BSA in PBS. The grids were then treated overnight (16–18 h) at 4 °C with a 1:20 dilution of a primary anti-Gpx-2 antibody in BSA. By incubating the sections for 1 h on immunogold conjugated goat anti-mouse IgG at 15 nm (BBInternational BBI Solutions, Sittingbourne, UK) diluted to 1:100, the bound antibodies were localized. Finally, before staining with 0.5% aqueous uranyl acetate, the grids were rinsed with PBS drops (five changes, 5 min each) and water (three changes, 3 min each). The main antibody was not used in the controls. The grids were then air-dried before being examined at 120 kV in a TECNAI 12 G2 Spirit Bio Twin FEI Company transmission electron microscope. A Morada CCD camera (Gatan RIO 9, Pleasanton, CA, USA) was used to collect the images.

### 4.4. Statistical Analysis

In the present study, an analysis of the relationship between the immunohistochemical expression of Gpx-2 and relevant clinical parameters was performed using the Statistica 9.1 software package developed by StatSoft, Krakow, Poland. The statistical measures of median and range were used to evaluate all numerical variables. Both the chi-squared test (Ch^2^ test) and Fisher’s exact test were used to assess the relative characteristics (treated as independent variables) of the groups studied. The aim of the present study was also to investigate the possible association between the intensity of Gpx-2 expression and patient survival. The relationship between the intensity of Gpx-2 expression and the 5-year survival of patients was tested using a Kaplan–Meier analysis and log-rank test. The results were considered to be statistically significant if *p* < 0.05.

## 5. Conclusions

Our preliminary report indicates that Gpx-2, a protein, is connected with reduced 5-year survival in colon adenocarcinoma patients. The multivariate analysis suggests that the grade of histological differentiation and immunohistochemical expression of Gpx-2 in colon adenocarcinoma tissue can be considered as an independent prognostic factor. It is worth noting that our study is the first to demonstrate the immunohistochemical expression of Gpx-2 in colon adenocarcinoma tissues in European populations. Additionally, it reveals the prognostic value of Gpx-2 expression in patients stratified by certain criteria that are relevant from a clinical perspective, such as the grade of histological differentiation, depth of invasion, staging and PCNA expression. Our work is also the first to demonstrate the localization of Gpx-2 in tumour tissue at the electron microscopic level using the immunogold labelling method.

However, we must acknowledge certain limitations in our study. The size of the cohort that we studied was small, and our patients were from only one hospital, which may have created a bias in our findings. To improve our understanding, future studies should aim to increase the sample size. Additionally, it would be interesting to determine the levels of Gpx-2 protein using the Western blot method in a larger group of patients, as well as to examine the mRNA content in tissue samples from both the tumour and the surrounding area without neoplastic lesions. In vitro studies, especially RNA-seq experiments, would be useful to investigate Gpx-2 activity at the subcellular level.

## Figures and Tables

**Figure 1 ijms-24-14650-f001:**
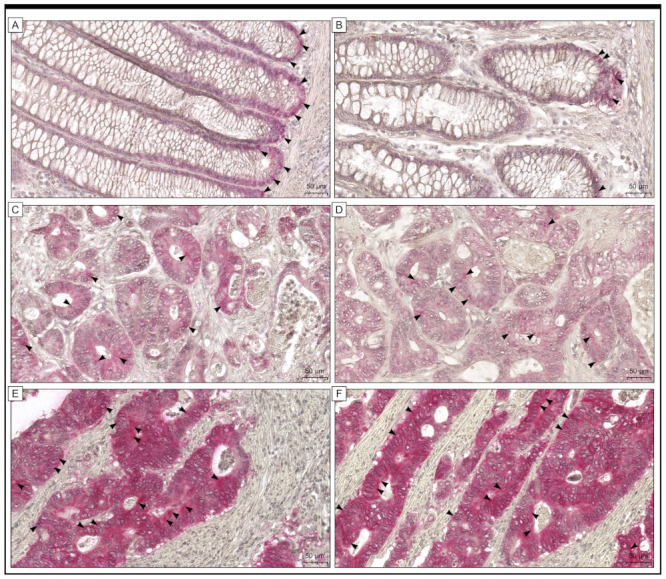
The expression of Gpx-2 in colon adenocarcinoma tissue (**C**–**F**) and adjacent non-cancerous tissue margins (**A**,**B**) is shown in this study with photomicrographs and black arrows. (**A**,**B**) Low levels of immunohistochemical reactivity were detected in cells found within the tissue margin of the colonic mucosa of non-pathological specimens. The expression was detected mainly in the base of crypts. In colon adenocarcinoma tissues, expression of Gpx-2 can be classified into two distinct groups, namely low (**C**,**D**) and high (**E**,**F**). The scale bar is 50 µm (**A**–**F**).

**Figure 2 ijms-24-14650-f002:**
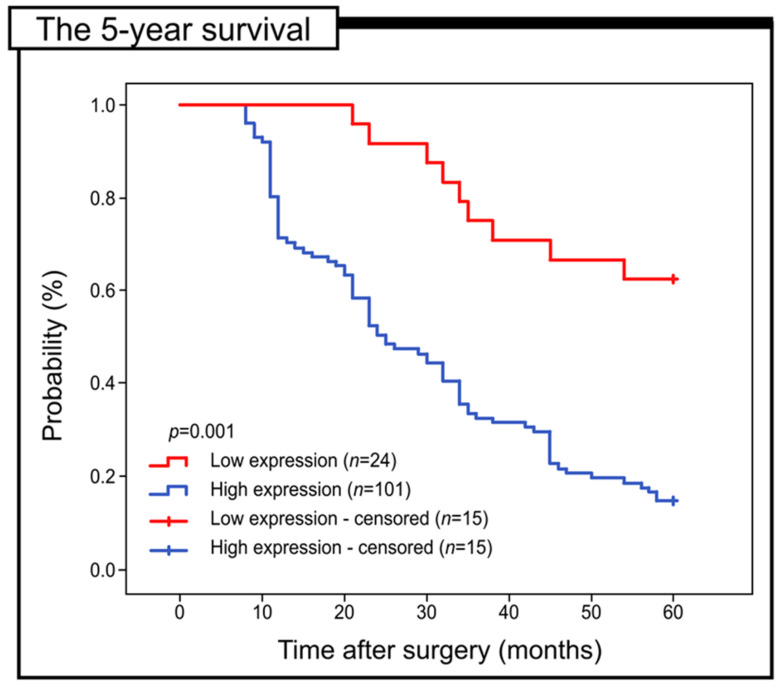
Kaplan–Meier curves of univariate analysis data (log-rank test) showing the 5-year survival rate for patients with high versus low Gpx-2 immunohistochemical expression.

**Figure 3 ijms-24-14650-f003:**
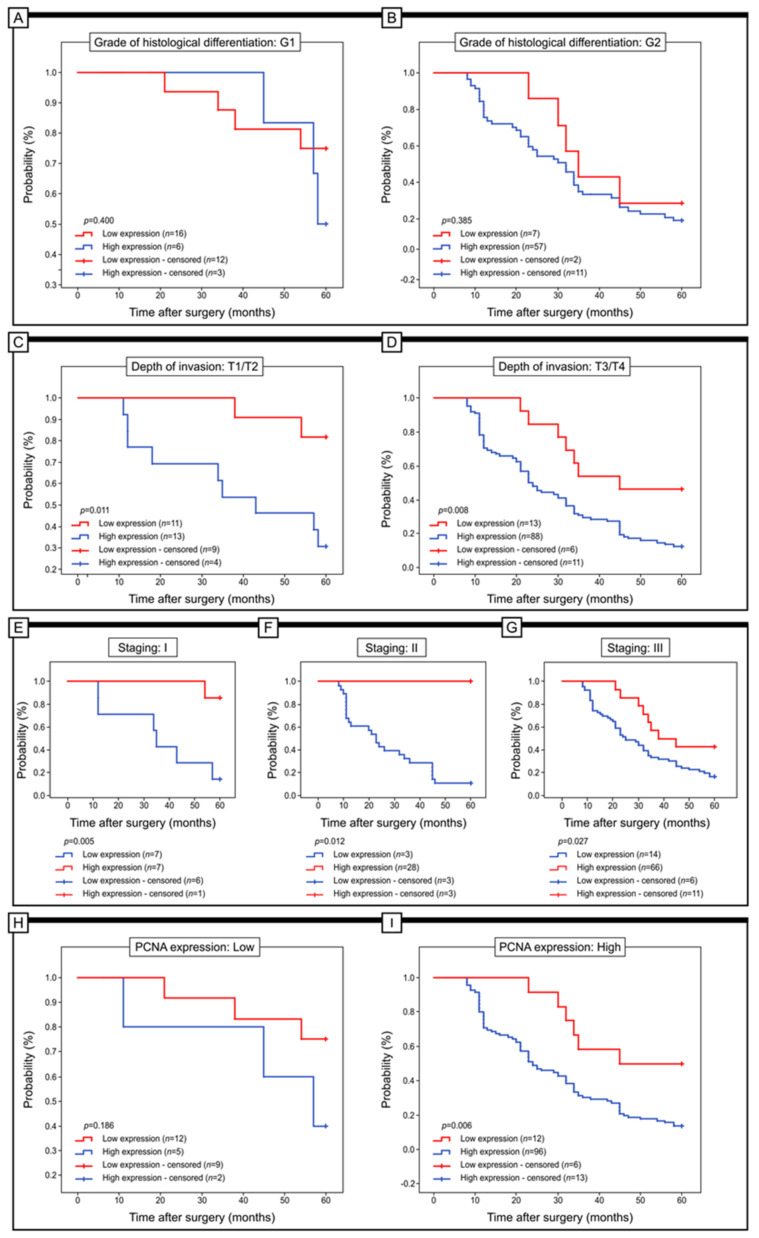
Kaplan—Meier curves of univariate analysis data (log—rank test) of patients with high versus low levels of Gpx-2 immunohistochemical expression. (**A**,**B**) Five-year survival of patients with G1 (**A**) and G2 grade of histological differentiation (**B**). Depth of invasion (**C**,**D**), 5-year survival of patients with T1/T2 (**C**) and T3/T4 (**D**) depth of invasion. (**E**,**F**) Five-year survival of patients with stage I (**E**), II (**F**) and III (**G**). (**H**,**I**) Five-year survival of patients with low (**H**) and high (**I**) expression of PCNA.

**Figure 4 ijms-24-14650-f004:**
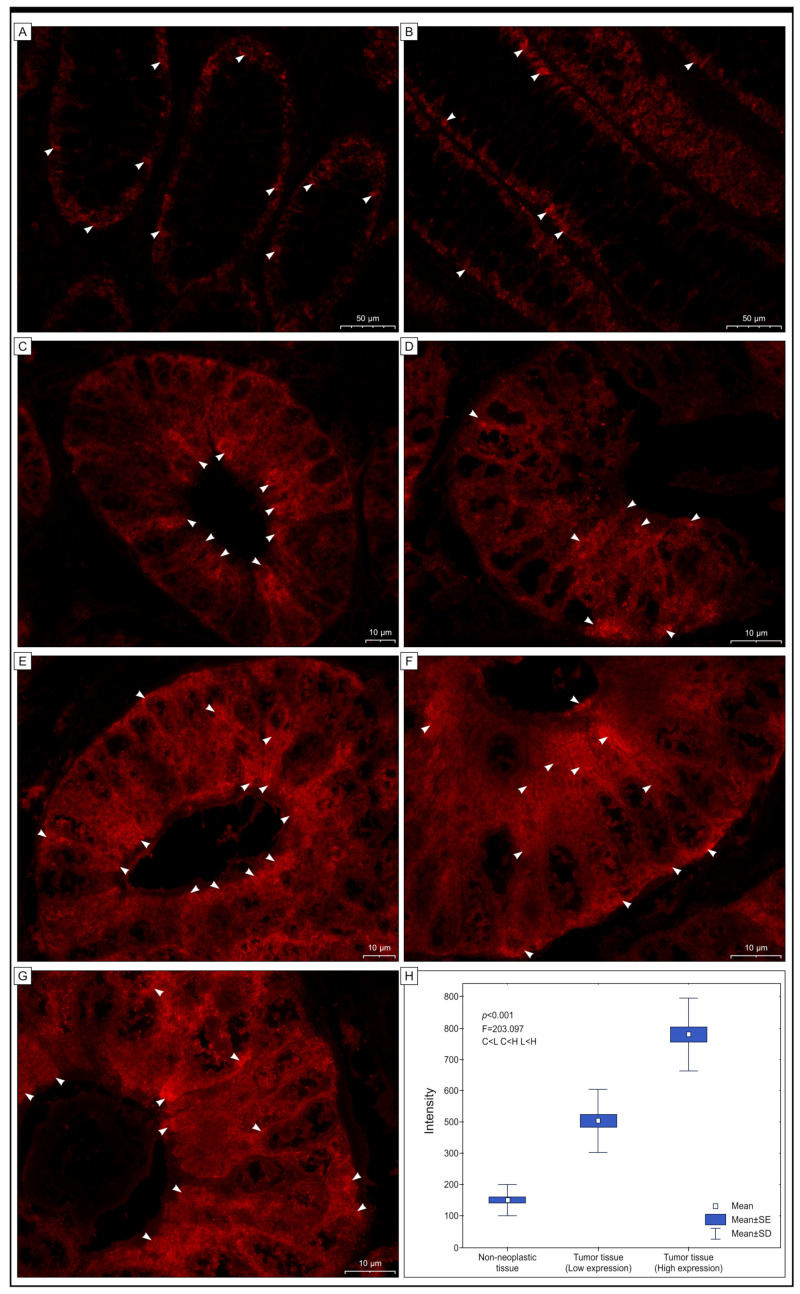
The images displayed here show the expression of Gpx-2 in colon adenocarcinoma tissue (**C**–**G**) and non-cancerous tissue margins (**A**,**B**) through immunofluorescence. A fluorescent signal (red colour signal) of varying intensity was found in cells of non-neoplastic mucosa (**A**,**B**)—arrowheads indicate expression in the cytoplasm of non-pathological colonocytes and cancer cells (**C**–**G**). In some cancer cells, the expression and fluorescence signal was found in the cytoplasm of the apical parts of the cells, while in others, intense fluorescence was found throughout all cytoplasm (arrowheads) of the cells or in the cell nuclei (arrows). (**H**) results of ANOVA test showing differences between the intensity of red signal indicating the presence of Gpx-2 among the tested groups; C < L, C < H, L < H—differences in intensity between the groups; C—non-neoplastic colon tissue, L—adenocarcinoma specimens with low expression of Gpx-2, H—adenocarcinoma specimens with high expression of Gpx-2.

**Figure 5 ijms-24-14650-f005:**
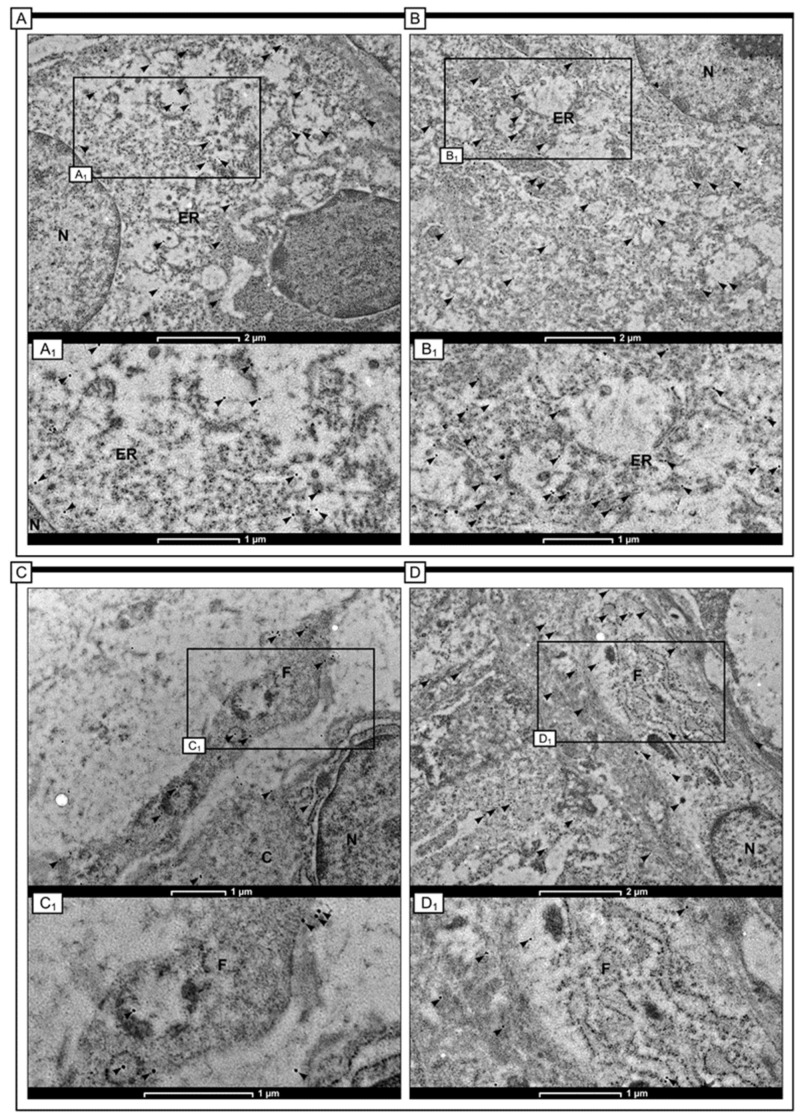
Detection of Gpx-2 protein with immunogold labelling in cells of colon adenocarcinoma tissue. The small electron-dense granules (arrowheads) were found in cancerous cells (**A**,**B**) and stromal cells, especially in fibroblast-like cells (**C**,**D**). In cancerous cells, the granules indicating the presence of Gpx-2 were detected within the cytoplasm and in this compartment, they were connected with the cisterns of the endoplasmic reticulum (**A_1_**,**B_1_**—black arrows). In fibroblasts, the localisation of Gpx-2 was connected with the cytoplasm as well. In this case, the gold granules were also detected within the membranous components in the cytoplasm. Moreover, the Gpx-2 in fibroblast cells was also associated with the plasma membrane (**C_1_**,**D_1_**—black arrows). Some cells also showed granules near the intermediate filament bundles. The scale bars are 1 µm (**A_1_**,**B_1_**,**C**,**C_1_**,**D_1_**) and 2 µm (**A**,**B**,**D**).

**Table 1 ijms-24-14650-t001:** Characteristics of patients included in the study (*n* = 125).

		*N* (Number of Cases)	%
Gender	Females	60	48.00
	Males	65	52.00
Age (years)	≤60 years	43	34.40
	61–75 years	44	35.20
	>75 years	38	30.40
	M ± SD	65.54 ± 13.11	
	Me (Q1–Q3)	65 (56–77)	
	Min–Max	33–89	
Grade of histological differentiation (G)	G1	22	17.60
	G2	64	51.20
	G3	39	31.20
Depth of invasion (T)	T1	10	8.00
	T2	14	11.20
	T3	78	62.40
	T4	23	18.40
Regional lymph node involvement (N)	N0	45	36.00
	N1	45	36.00
	N2	35	28.00
Location of tumour	Proximal	63	50.40
	Distal	62	49.60
Angioinvasion	No	27	21.60
	Yes	98	78.40
PCNA immunohistochemical expression	Low	17	13.60
	High	108	86.40
Staging	I	14	11.20
	II	31	24.80
	III	80	64.00

**Table 2 ijms-24-14650-t002:** Correlations between the expression of Gpx-2 protein and PCNA protein (as dependent variables).

		The Immunoexpression Level of Gpx-2				*p*-Value
		Low		High		
PCNA expression	Low	12	(9.60%)	5	(4.00%)	*p* < 0.001
	High	12	(9.60%)	96	(76.80%)	*p* = 0.146

**Table 3 ijms-24-14650-t003:** Correlations between the expression of Gpx-2 protein and clinicopathological characteristics in colon adenocarcinoma patients (as independent variables).

		The Immunoexpression Level of Gpx-2				
		Low		High		*p*-Value
Age (years)	≤60 years	7	(16.28%)	36	(83.72%)	*p* = 0.410
	61–75 years	7	(15.91%)	37	(84.09%)	
	>75 years	10	(26.32%)	28	(73.68%)	
Gender	Females	13	(21.67%)	47	(78.33%)	*p* = 0.501
	Males	11	(16.92%)	54	(83.08%)	
Grade of histological differentiation (G)	G1	16	(72.73%)	6	(27.27%)	*p* < 0.001
	G2	7	(10.94%)	57	(89.06%)	
	G3	1	(2.6%)	38	(97.44%)	
Depth of invasion (T)	T1/T2	11	(45.83%)	13	(54.17%)	*p* = 0.001
	T3/T4	13	(12.87%)	88	(87.13%)	
Regional lymph node involvement (N)	N0	10	(22.22%)	35	(77.78%)	*p* = 0.661
	N1	9	(20.00%)	36	(80.00%)	
	N2	5	(14.29%)	30	(85.71%)	
Location of tumour	Proximal	8	(12.70%)	55	(87.30%)	*p* = 0.063
	Distal	16	(25.81%)	46	(74.19%)	
Angioinvasion	No	15	(55.56%)	12	(44.44%)	*p* < 0.001
	Yes	9	(9.18%)	89	(90.82%)	
PCNA immunohistochemical expression	Low	12	(70.59%)	5	(29.41%)	*p* < 0.001
	High	12	(11.11%)	96	(88.89%)	
Staging	I	7	(50.00%)	7	(50.00%)	*p* = 0.005
	II	3	(9.68%)	28	(90.32%)	
	III	14	(17.50%)	66	(82.50%)	

**Table 4 ijms-24-14650-t004:** Univariate and multivariate analyses of various prognostic parameters in colon adenocarcinoma patients using Cox regression analyses.

Prognostic Parameter	Univariate Analysis			Multivariate Analysis		
	HR	95% CI	*p*-Value	HR	95% CI	*p*-Value
Gender	0.898	0.600–1.343	0.600	-	-	**-**
Age	0.999	0.985–1.014	0.912	-	-	**-**
Grade of histological differentiation (G)	2.382	1.751–3.241	<0.001	1.919	1.254–2.936	0.003
Depth of invasion (T)	1.664	1.265–2.188	<0.001	1.045	0.739–1.478	0.803
Regional LN involvement (N)	1.136	0.885–1.458	0.318	-	-	**-**
Location of tumour	1.063	0.711–1.590	0.767	-	-	**-**
Gpx-2 immunohistochemical expression	4.046	2.028–8.072	<0.001	2.286	1.069–4.890	0.033
Angioinvasion	2.745	1.522–4.950	0.001	0.948	0.434–2.070	0.893
PCNA immunohistochemical expression	4.133	1.800–9.486	0.001	1.242	0.387–3.987	0.715
Staging	1.287	0.956–1.731	0.096	-	-	-

## Data Availability

All data generated or analysed during this study are included in this article. Further inquiries can be directed to the corresponding author.

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
