# Peer review of "Immunohistochemical Expression of Glutathione Peroxidase-2 (Gpx-2) and Its Clinical Relevance in Colon Adenocarcinoma Patients"

_ijms, 2023, doi:10.3390/ijms241914650_

Round 1

Reviewer 1 Report

This article titled " Immunohistochemical Expression of Glutathione Peroxidase-2 (Gpx-2) and Its Clinical Relevance in Colon Adenocarcinoma Patients" provides valuable insights into the Gpx-2 protein expression in 65 patients with colorectal adenocarcinoma who underwent radical surgery without prior treatment. Correlations between Gpx-2 expression and clinical factors were explored, revealing associations with tumor grade, PCNA expression, invasion depth, and angioinvasion. Notably, high Gpx-2 expression was linked to poorer 5-year survival rates, suggesting its potential as a prognostic marker for colon adenocarcinoma.

Below concerns should be address in a revised manuscript before acceptance for publication.

Comments:

  1. Paper needs to be put in better context with prior studies and current study in order to make readers easily distinguished what is already known, and what is truly novel about your study. This point is very important for presenting the novelty and significance of current study.
  2. Current study found that Gpx-2 protein highly expressed in colorectal adenocarcinoma who underwent radical surgery without prior treatment. I think western blot experiment is required to confirm these results and provide more solid evidence to support authors’ point.
  3. RNA-seq experiment will be a good strategy to find the relationship and regulation mechanism of Gpx-2 and tumor development.
  4. It would be beneficial to include more discussion on the implications of the findings and their potential applications in biomarker and drug discovery.
  5. The manuscript still requires some editing work with respect to grammar and sentence construction.

The manuscript still requires some editing work with respect to grammar and sentence construction.

Author Response

Dear Reviewer,

We are extremely grateful for all critical remarks concerning our work indicating your very profound analysis of our manuscript

  1. Paper needs to be put in better context with prior studies and current study in order to make readers easily distinguished what is already known, and what is truly novel about your study. This point is very important for presenting the novelty and significance of the current study.

Thank you very much for this suggestion. In accordance with the reviewer's suggestion, the introduction has been amended to clarify what is “new” in my study compared to others. It has been made clear that there is a lack of data on the expression of Gpx-2 in patients with colon and rectal cancer. The clinical and prognostic reference of this protein in colon adenocarcinoma patients is also missing. Our study clearly fills these gaps and focuses on a group of patients of European origin.

  1. Current study found that Gpx-2 protein highly expressed in colorectal adenocarcinoma who underwent radical surgery without prior treatment. I think western blot experiment is required to confirm these results and provide more solid evidence to support authors’ point.

I am very grateful to the reviewer for this valuable suggestion. In our current study, we decided to focus on immunohistochemical evaluation of Gpx-2 protein expression, which is a very useful technique in clinical application and also for oncologists. We decided to confirm our results using fluorescence microscopy and transmission electron microscopy. Unfortunately, we cannot do any more additional research because the research budget has been closed and there is no more money to buy reagents. However, we plan to continue working on the role of the Gpx-2 protein in colon adenocarcinoma samples. Therefore, we have included this suggestion in the next draft. We are very grateful for this suggestion. Our plan is to look at the levels of the protein in colon adenocarcinoma tissues and in cells lines. We hope to get funding for this.

RNA-seq experiment will be a good strategy to find the relationship and regulation mechanism of Gpx-2 and tumor development.

I agree that this experiment is recommendable. Unfortunately, as I wrote earlier, we cannot do any more research because the research budget has been closed. We have just included the RNA-seq experiment in our next project. We are very grateful for this suggestion and we will include this point at the end of the discussion where we explain the limitations of our research.

It would be beneficial to include more discussion on the implications of the findings and their potential applications in biomarker and drug discovery.

Thank you very much for this suggestion. According to the Reviewer's suggestion, I changed the part of the discussion and I included the implications of my finding and their potential application.

The manuscript still requires some editing work with respect to grammar and sentence construction.

Thank you very much for your opinion. I did everything I could in terms of editing and English improvement.

Reviewer 2 Report

Immunohistochemical Expression of Glutathione Peroxidase-2 (Gpx-2) and Its Clinical Relevance in Colon Adenocarcinoma Patients

GENERAL COMMENTS: In this article, the authors evaluated the expression of Gpx-2 protein (using the immunohistochemistry) in colon adenocarcinoma tissues as well as its intracellular localization. Furthermore, they analyzed the correlation of Gpx-2 expression with other clinicopathological characteristics and determined the prognostic significance of Gpx-2 expression in these tumors. 

On the plus side, the authors perform a number of experiments that have adequate controls, statistical significance. Although the authors demonstrated the usefulness of Gpx-2 as a prognostic marker in colon adenocarcinomas, the article's negatives are unclear wording, many errors and inaccuracies.

Major remarks:

-        The article is focused on colon adenocarcinoma, but in some places, line 150, 186 and 338, colorectal adenocarcinoma is mentioned. These two terms are not synonymous.

-        Abbreviations section is missing. Some abbreviations are not explained in the text where they first appeared.

-        English needs to be thoroughly revised. In some sentences, the word order is strange and the sentences are difficult to understand.

-        The text would be much clearer and easier to understand if references to literature or to figures, graphs and tables were located closest to the place where they are first mentioned (eg at the end of the relevant sentence).

-        Decimal points are given sometimes as dots, sometimes as commas.

List of shortcomings / errors that should be corrected / clarified:

Materials and Methods

1.       lines 345 – 346: The authors state that “To evaluate the prognostic significance of the URG4 protein, …..” - The main goal of this research was to determine the prognostic significance of Gpx-2 in colon adenocarcinoma, as evidenced by the Kaplan Meier survival graphs in fig. 3 and 4. This sentence needs to be reformulated and supplemented with the correct proteins whose expression was prognostically evaluated.

2.       lines 355 – 356: the catalog numbers of anti-Gpx-2 and PCNA antibodies are the same.

3.       lines 364 – 365: I recommend rephrasing this sentence: “The score was based on both the intensity and extension of the immunohistochemical reaction, and it determined the presence of Gpx-2.” The term "extension of the immunohistochemical reaction" does not describe the evaluated parameter - the number of positive cells.

4.       Line 376 - the entire paragraph needs to be revised - it makes no sense. In the first sentence, anti Gpx-2 antibody is mentioned, but in the next sentence it is written about the evaluation of Notch4 expression.

5.       Line 384 - it is necessary to specify which samples were used for electron microscopy.

Results

1.       Figure 1 - The scale bar is not visible.

2.       Figure 2 - This figure is actually just a graphic representation of Table 2 - the distribution of patients based on the combined analysis of low and high expression of PCNA and Gpx-2, and therefore has no significant informative value for the reader.  Moreover, figure description is imprecise and insufficient.

3.       Lines 136 and 160 - URG4 expression levels is discussed. Is the expression of the URG 4 protein also evaluated? How? With what results? If so, what is the relationship between URG4 and Gpx-2 expression?

4.       Line 188: The text contains a citation of an incorrect table. Table 4 should be listed there correctly.

Statistics

Yates’ correction was used previously to provide a more conservative result for contingency tables with small cell counts. Currently, Fisher's exact test provides a better solution to dealing with small cell counts and is preferred.

Author Response

Dear Reviewer,

We are extremely grateful for all critical remarks concerning our work indicating your very profound analysis of our manuscript

  1. The article is focused on colon adenocarcinoma, but in some places, line 150, 186 and 338, colorectal adenocarcinoma is mentioned. These two terms are not synonymous.

Thank you for this valuable comment. We have of course made the appropriate correction. We have also defined the term colorectal cancer (CRC) as cancer of the colon and rectum. This is a term that has been used by researchers. However, it is not entirely appropriate, as we have also mentioned. Thank you again for your valuable comment. In our work, we use the term colon adenocarcinoma because we have focused on this malignancy.

  1. Abbreviations section is missing. Some abbreviations are not explained in the text where they first appeared.

Thank you for your comment. We have, of course, added a section on abbreviations and explained in the main text all abbreviations used for the first time.

English needs to be thoroughly revised. In some sentences, the word order is strange and the sentences are difficult to understand.

Thank you for your comment. English was revised.

The text would be much clearer and easier to understand if references to literature or to figures, graphs and tables were located closest to the place where they are first mentioned (eg at the end of the relevant sentence). Decimal points are given sometimes as dots, sometimes as commas.

In accordance with the reviewer's suggestion, we have made the necessary corrections

Materials and Methods

  1. lines 345 – 346: The authors state that “To evaluate the prognostic significance of the URG4 protein, …..” - The main goal of this research was to determine the prognostic significance of Gpx-2 in colon adenocarcinoma, as evidenced by the Kaplan Meier survival graphs in fig. 3 and 4. This sentence needs to be reformulated and supplemented with the correct proteins whose expression was prognostically evaluated.

Thank you for your comments. We are very sorry for this error, which was due to our lack of attention. The expression of Gpx-2 is, of course, what we have in mind.

  1. lines 355 – 356: the catalog numbers of anti-Gpx-2 and PCNA antibodies are the same.

We have made the necessary corrections

  1. lines 364 – 365: I recommend rephrasing this sentence: “The score was based on both the intensity and extension of the immunohistochemical reaction, and it determined the presence of Gpx-2.” The term "extension of the immunohistochemical reaction" does not describe the evaluated parameter - the number of positive cells.

Thank you for your comments. In accordance with the reviewer's suggestion, we have made the necessary corrections.

  1. Line 376 - the entire paragraph needs to be revised - it makes no sense. In the first sentence, anti Gpx-2 antibody is mentioned, but in the next sentence it is written about the evaluation of Notch4 expression.

Thank you for your comments. In accordance with the reviewer's suggestion, we have made the necessary corrections. The expression of Gpx-2 is, of course, what we have in mind.

  1. Line 384 - it is necessary to specify which samples were used for electron microscopy.

Thank you for your comments. The correction has been made.

Results

  1. Figure 1 - The scale bar is not visible.

We have made the change so that the scale bar is now visible in the image.

  1. Figure 2 - This figure is actually just a graphic representation of Table 2 - the distribution of patients based on the combined analysis of low and high expression of PCNA and Gpx-2, and therefore has no significant informative value for the reader. Moreover, figure description is imprecise and insufficient.

We agree that this figure has no significant information value for the reader. We have therefore decided to remove it.

  1. Lines 136 and 160 - URG4 expression levels is discussed. Is the expression of the URG 4 protein also evaluated? How? With what results? If so, what is the relationship between URG4 and Gpx-2 expression?

Thank you for your comments. In accordance with the reviewer's suggestion, we have made the necessary corrections. The expression of Gpx-2 is, of course, what we have in mind.

  1. Line 188: The text contains a citation of an incorrect table. Table 4 should be listed there correctly.

Thank you for your comments. In accordance with the reviewer's suggestion, we have made the necessary corrections.

Yates’ correction was used previously to provide a more conservative result for contingency tables with small cell counts. Currently, Fisher's exact test provides a better solution to dealing with small cell counts and is preferred.

Thank you for the advice. We have replaced the Yates correction with Fisher's test.